# Reliability and Validity of the CORE Sensor to Assess Core Body Temperature during Cycling Exercise

**DOI:** 10.3390/s21175932

**Published:** 2021-09-03

**Authors:** Nina Verdel, Tim Podlogar, Urša Ciuha, Hans-Christer Holmberg, Tadej Debevec, Matej Supej

**Affiliations:** 1Swedish Winter Sports Research Centre, Mid Sweden University, 83125 Östersund, Sweden; matej.supej@fsp.uni-lj.si; 2Faculty of Health Sciences, University of Primorska, 6310 Izola, Slovenia; tim@kineziolog.si; 3Department of Automation, Biocybernetics, and Robotics, Jozef Stefan Institute, 1000 Ljubljana, Slovenia; ursa.ciuha@ijs.si (U.C.); tadej.debevec@fsp.uni-lj.si (T.D.); 4Human Performance Centre, 1000 Ljubljana, Slovenia; 5Department of Health, Medicine, and Rehabilitation, Luleå University of Technology, 97187 Luleå, Sweden; integrativephysiobiomech@gmail.com; 6Faculty of Sport, University of Ljubljana, 1000 Ljubljana, Slovenia

**Keywords:** validity, reliability, core body temperature, rectal temperature, CORE sensor, cycling, non-invasive

## Abstract

Monitoring core body temperature (*T*_c_) during training and competitions, especially in a hot environment, can help enhance an athlete’s performance, as well as lower the risk for heat stroke. Accordingly, a noninvasive sensor that allows reliable monitoring of *T*_c_ would be highly beneficial in this context. One such novel non-invasive sensor was recently introduced onto the market (CORE, greenTEG, Rümlang, Switzerland), but, to our knowledge, a validation study of this device has not yet been reported. Therefore, the purpose of this study was to evaluate the validity and reliability of the CORE sensor. In Study I, 12 males were subjected to a low-to-moderate heat load by performing, on two separate occasions several days apart, two identical 60-min bouts of steady-state cycling in the laboratory at 19 °C and 30% relative humidity. In Study II, 13 males were subjected to moderate-to-high heat load by performing 90 min of cycling in the laboratory at 31 °C and 39% relative humidity. In both cases the core body temperatures indicated by the CORE sensor were compared to the corresponding values obtained using a rectal sensor (*T*_rec_). The first major finding was that the reliability of the CORE sensor is acceptable, since the mean bias between the two identical trials of exercise (0.02 °C) was not statistically significant. However, under both levels of heat load, the body temperature indicated by the CORE sensor did not agree well with *T*_rec_, with approximately 50% of all paired measurements differing by more than the predefined threshold for validity of ≤0.3 °C. In conclusion, the results obtained do not support the manufacturer’s claim that the CORE sensor provides a valid measure of core body temperature.

## 1. Introduction

During the 2020 Olympic Games in Tokyo and 2022 FIFA World Cup in Qatar, temperatures above 30 °C are expected [1,2]. Prolonged, intense exercise in such a hot environment impairs athletic performance [3], causes a rise in core body temperature (*T*_c_) and increases the risk for potentially life-threatening exertional heat illness (heat stroke) associated with a *T*_c_ above 40 °C [4]. To offset the impact of thermally stressful environmental conditions, numerous strategies have been developed to help manage heat stress. Out of these strategies, heat acclimation and heat acclimatization appear to provide optimal benefits [5]. For optimal heat acclimation, the training guidelines advise athletes to exercise for a prolonged time (e.g., 60–90 min) at *T*_c_ above 38.5 °C [6,7]. Monitoring *T*_c_ in training is, therefore, an important part of the training process both for achieving the desired stimuli/adaptation to a given training session and secondly to prevent heat-related medical issues. It is therefore of utmost importance to provide athletes and coaches with a valid, reliable, and easily applicable strategy to monitor *T*_c_.

*T*_c_ can be assessed at different body sites, such as the rectum, esophagus, pulmonary artery, mouth, aural canal, armpit, and forehead. For a comprehensive overview of different *T*_c_ measurement methodologies, the readers are recommended to consult a recent review by Tyler et al. [8]. With respect to validity, measurements in the mouth, aural canal, and armpit, and on the forehead remain questionable [9], while measurements of *T*_c_ in the pulmonary artery, esophagus, and the rectum have been shown to provide valid *T*_c_ data. However, measurement of the temperature in the pulmonary artery or esophagus is invasive, requires trained medical personnel, and thus remains of limited use, even in laboratory settings [10]. On the other hand, measurement of rectal temperature (*T*_rec_) represents a valid and reliable *T*_c_ measurement for individuals at rest and while exercising [11,12] and is employed in the majority of sports science thermoregulatory studies. In addition, *T*_rec_ also serves as the criterion standard for temperature measurement in hyperthermic athletes [13,14]. Despite its widespread use, the measurement of *T*_rec_ has several important limitations. Prolonged sitting with the inserted rectal probe may be uncomfortable for athletes, the measurement is mostly limited to the laboratory conditions, due to body movement the sensor can be displaced from its original position, as well as the movement of the hips may be limited [15]. Therefore, the measurement of *T*_rec_ during training (heat acclimatization) or competitions remains inconvenient. 

In recent decades ingestible temperature sensors (pills) became a popular alternative for research and professional sport. Several studies provide evidence to suggest that ingestible pills are valid sensors for the assessment of *T*_c_ [15,16,17]. However, this technique also has several limitations, including the fact that the pill has to be ingested a few hours before the exercise, it can become contaminated by food or fluid ingestion, and it is expensive as well. Therefore, monitoring of the *T*_c_ during each training session or competition with ingestible pills is not widespread. 

Hence, a potential noninvasive sensor that would allow monitoring of *T*_c_ during each training or competition would provide significant benefits in terms of heat acclimation/acclimatization training and prevent the occurrence of a heatstroke. Clearly, a noninvasive sensor that would allow accurate monitoring of *T*_c_ during specific periods would be of considerable benefit not only to athletes, but also for workers exposed to high thermal loads (e.g., firemen and soldiers), as well as for obtaining important diagnostic information in clinical settings [18].

Recently, one such sensor, the CORE (greenTEG AG, Rümlang, Switzerland), has become commercially available [19]. The CORE apparatus involves a novel type of thermal energy transfer sensor (a heat flux sensor) that determines core temperature using machine learning algorithms based on measurements of heat flux and skin temperature, as well as, when exercising, data provided by external heart rate sensors connected to the CORE via ANT+ protocol. Although the CORE sensor is already used by many athletes, even in world-class competitions (such as the cycling Tour de France), to our knowledge, a peer-reviewed validation study has not yet been published. Unfortunately, lately, different wearables such as the CORE sensor are often marketed with aggressive and potentially exaggerated claims that lack a sound scientific basis [20,21]. Accordingly, the present investigation was designed to compare the validity and reliability of the CORE sensor to that of a rectal sensor under various laboratory conditions, as well as to examine the reproducibility of values obtained with the CORE sensor during exercise under the same conditions on two separate occasions.

## 2. Materials and Methods

### 2.1. Study Design

Twenty-four healthy and physically active male volunteers (age = 30 ± 5 years; body mass = 77.9 ± 9.6 kg; height = 180 ± 7 cm; peak oxygen uptake($\dot{𝑉}$*O*_2_peak) = 58 ± 7 mL min^−1^ kg^−1^ (means ± standard deviations)) participated in this study, which was approved by the Ethics Committee for sport at the University of Ljubljana, Slovenia (033-3/2021-2), which adheres to the principles outlined by the World Medical Assembly Declaration of Helsinki. Informed consent was obtained from all subjects involved in the study. The subjects included were men younger than 40 years of age who cycled for at least 8 h each week. 

### 2.2. Design of Studies I and II

Study I was designed to assess the reliability and validity of the CORE sensor with low-to-moderate heat load, while Study II was designed to assess the validity of the CORE sensor under moderate-to-high heat load. For these evaluations, the participants came to the laboratory on three and two separate occasions, respectively. In both cases, the participants underwent a pretest in connection with the first visit. The protocol for Trials 1a and 1b performed during the subsequent two visits to the laboratory in Study I were identical, allowing us to determine the test-retest reliability of the CORE sensor.

Upon each arrival, participants’ body mass was measured and the participants were asked to insert a rectal probe 12 cm past the anal sphincter in a private room. Additionally, each participant was equipped with a heart rate chest strap with an attached CORE sensor (explained in detail below). All exercise trials were performed using participants’ own bicycles mounted on an electrically braked cycle ergometer (Kickr V5, Wahoo, Atlanta, GA, USA). 

#### 2.2.1. Pretest

The pretest visit with the aim to assess the participants’ baseline characteristics, i.e., the peak oxygen uptake, and associated power output (*W*_max_), was the same for both studies. Additionally, exercise intensities corresponding to the first ventilatory threshold (VT1), and the respiratory compensation point (RCP) were determined as previously described by Iannetta et al. [22]. Ambient conditions were kept thermoneutral with temperature and relative humidity levels at 21.9 ± 0.2 °C and 36 ± 1%, respectively. 

In brief, the exercise test began with an 8-min warm-up: a 2-min at 80 W, followed by 6-min at 120 W. This was followed by an incremental ramp test to maximal volitional exertion. The intensity of the exercise was gradually increased by 30 W min^−1^ in a stepped manner. Pedaling frequency was self-selected, and participants were encouraged to continue until task failure. After 30 min of passive rest, participants cycled for 10 min at 50–65% W_peak_ (i.e., cycling in the heavy exercise intensity domain) to obtain the parameters required for the determination of exercise intensities corresponding to VT1 and respiratory compensation point (RCP).

During this test, gas exchange was monitored by an automated online system (MetaLyzer 3B-3R, Cortex, Biophysics GmbH, Leipzig, Germany). Before each trial gas analyzers were calibrated with a known gas mixture (15.10% O_2_, 5.06 CO_2_; Linde Gas A.S., Prague, Czech Republic), and the volume transducer was calibrated with a 3-L syringe (Cortex, Leipzig, Germany). Peak oxygen uptake was calculated as the highest 30-s average value of O_2_ consumption. 

#### 2.2.2. Study I

The 12 men (29 ± 5 years, 78.6 ± 10.2 kg, 181 ± 6 cm) who participated in the first study demonstrated a mean $\dot{𝑉}$*O*_2_ peak of 57.3 ± 6.4 mL kg^−1^ min^- 1^ and a *W*_max_ of 413 ± 49 W. The ambient conditions (laboratory temperature and relative humidity) during both trials were similar 19.1 ± 0.6 °C, 33 ± 7%, and 19.1 ± 0.5 °C, 32 ± 5%, respectively.

After the pretest participants were asked to visit the laboratory on two additional separate occasions (Trial 1a and Trial 1b) taking place at the same time of the day. They entered the laboratory after an overnight fast and having abstained from performing exercise 24 h before each trial. Additionally, participants recorded their diet 24 h before Trial 1a and replicated their diet for Trial 1b.The protocol began with cycling for 5 min at 60% VT1, followed by 60 min of steady-state exercise (SS) at 90% VT1.

#### 2.2.3. Study II

The 13 participants (31 ± 5 years, 178 ± 8 cm, 77.0 ± 9.0 kg) demonstrated a mean $\dot{𝑉}$*O*_2_peak of 59.0 ± 8.9 mL kg^−1^ min^- 1^, corresponding to a mean peak power of 410 ± 60 W. The ambient conditions (laboratory temperature and relative humidity) were 30.7 ± 0.7 °C, and 39.0 ± 6.0%, respectively.

The exercise started with a 5-min warm-up of cycling at an intensity of 100 W, followed by a 10-min of exercise with graded increases in power output corresponding to RCP in order to increase the heat production. Thereafter, the participants cycled for 60 min at SS intensity, with a subsequent 15 min of cooling down. The SS intensity was reduced if the thermal discomfort or *T*_rec_ of the participants was too high (*T*_rec_ above 39.5 °C). Prior to the arrival at the laboratory, participants were instructed to drink enough liquids and during the exercise session, drinks were provided ad libitum. 

### 2.3. Measurement of Temperature and Heart Rate

#### 2.3.1. Body Temperature

Core body temperature was measured with a MSR rectal sensor (MSR, Seuzach, Switzerland) as a reference and the greenTEG CORE sensor. The CORE wearable sensor (4 cm × 5 cm × 0.8 cm) estimates *T*_c_ based on the measurements of skin temperature, heat flux, and heart rate (optional). According to the manufacturer’s instructions this sensor must be positioned on the torso/chest approximately 20 cm below the armpit using a heart rate monitor strap. For measurements during physical activity, the CORE should be paired with the heart rate monitor (HRM), but this is not necessary otherwise. The manufacturer offers two different versions of this sensor: CORE and COREresearch, the latter of which samples data every second, can store this data for 3.5 days and was employed here. Data stored on the device can be downloaded to the Android or iOS CORE app for further analysis. The accuracy of the CORE device is described by the manufacturer to be ± 0.26 °C.

#### 2.3.2. Rectal Temperature

*T*_rec_ was determined with a rectal sensor connected to a data logger (MSR145WD, Seuzach, Switzerland) from which the data collected were later transfered to a personal computer via a USB. The accuracy of the MSR sensor is reported by the manufacturer to be ± 0.20 °C. 

#### 2.3.3. Heart Rate

Heart rate was measured with a Polar H10 heart rate sensor (Polar OY, Kempele, Finland) connected to the CORE sensor. 

### 2.4. Data Analysis

The acquisition frequency was 1 Hz for the CORE sensor and 0.1 Hz for the rectal MSR sensor. Therefore, averages per 10 s were calculated for the CORE sensor. These values have been used for statistical analysis performed with Matlab R2020b (MathWorks Inc., Natick, MA, USA).

### 2.5. Statistics 

The data were tested for normality by the Kolmogorov-Smirnov test as well as the differences between data. Because normality was rejected for all the data (*p* < 0.05), statistical tests that do not assume normality were used. 

#### 2.5.1. Reliability of the Device for Measuring the T_c_

Device measurements that were performed twice (Study I -Trials 1a and 1b) were evaluated for intra-device reliability. The Wilcoxon signed-rank test was used to assess the systematic bias between trials, with the statistical significance set at *p* < 0.05. Limits of agreement (LoA) were calculated according to a nonparametric approach, as proposed by Bland and Altman [23]. Briefly, values that fell outside 10% of the observations were identified and then 5% of the observations from each end were removed. In addition, the peak temperature values of both trials and the largest differences at a discrete-time point were compared with the Wilcoxon signed-rank test. Values are expressed as means ± standard deviations (SD). Ambient conditions data were normally distributed. Therefore, a paired *t*-test was used to assess between trial differences in ambient conditions. Statistical significance was set at *p* < 0.05.

#### 2.5.2. Validity of the Device for Measuring the *T_c_*

Validity was assessed by evaluating the association between the data provided by the CORE sensor and a rectal sensor. The concurrent validity, which evaluates the association between data provided by the new device (i.e., CORE) and another device considered to be more valid (i.e., rectal sensor), is reported. The temperature device validity statistics were similar to those described in Section 2.5.1. (i.e., bias, limits of agreement, peak values, maximal differences). The acceptable difference between devices was taken as ≤ 0.3 °C [9,15,16].

## 3. Results

### 3.1. Reliability

The first study—Study I—involved two visits (Trial 1a and Trial 1b), both with the same protocol. There were no statistically significant differences for laboratory ambient temperature (*p* = 0.95) and room relative humidity (*p* = 0.76) between Trial 1a (19.1 ± 0.6 °C, 33 ± 7%) and Trial 1b (19.1 ± 0.5 °C, 32 ± 5%), respectively. 

Figure 1a presents the mean CORE temperature of Trials 1a (pink line) and 1b (violet line) with the corresponding standard deviations (pink and violet shaded area). There was no statistically significant difference in mean CORE temperature between both trials (0.02 ± 0.23 °C, *p* = 0.622), see Table 1. 

The analysis was further performed separately for the warm-up period and every 20 min of SS. Wilcoxon signed-rank test showed that the mean temperature difference was statistically significant for the warm-up period (0.25 ± 0.34, *p* = 0.027), while there was no statistically significant difference between the mean temperature of both trials for either part of the SS, Table 1.

A Bland Altman plot is presented in Figure 1b. The limits of agreement for the whole workout are −0.30 °C and 0.42 °C, see Table 1. In addition, the analysis showed the minimum range between LoA for the second 20 min of SS and the largest for the warm-up period (−0.16 to 0.21 °C, and −0.34 to 0.86 °C).

Table 2 presents the mean temperatures and temperature increase of the participants of the entire exercise bout and at the end of each phase (warm-up; initial, middle, and final 20 min of SS). There were no statistically significant differences in temperature between Trials 1a and 1b at the end of each phase. The temperature increase was significantly lower for the entire exercise and the initial 20 min of SS during Trial 1b (*p* = 0.021, and *p* = 0.043), while there were no statistically significant differences in temperature rise for other phases of the exercise.

### 3.2. Validity

Validity was assessed by utilizing two different protocols. In the first protocol, participants were exposed to low-to-moderate heat load, and in the second protocol to moderate-to-high heat load. 

#### 3.2.1. Low-to-Moderate Heat Load

Validity at low-to-moderate heat load was tested on the same participants as in Section 3.1, and the protocol was the same as well. The ambient temperature was 19.1 ± 0.6 °C, and the humidity 33 ± 7%. 

Figure 2a presents the mean *T*_c_ of the participants, measured with the MSR rectal sensor and the CORE sensor with corresponding standard deviations. The difference in mean core body temperature measured with the MSR rectal sensor and the CORE sensor was statistically significant (0.23 ± 0.35 °C, *p* < 0.001), see Figure 2. Mean core body temperature (with standard deviations) as determined with the MSR rectal sensor (red line and shaded area) and CORE (blue) sensor during Study I. (**b**) Bland-Altman plot of the data in (**a**). The dashed lines represent the limits of agreement (LoA) and bias for the entire period of exercise.

The positive mean bias means that the CORE sensor overestimated the temperature from the rectal sensor. Moreover, data show that mean differences between devices were below *a prior* established threshold of 0.3 °C threshold in 51% of all values for the entire exercise, see Table 3. 

The analysis was performed separately for the warm-up period and every 20 min of SS. Wilcoxon signed-rank test showed that the mean temperature difference was statistically significantly different for the warm-up period (0.25 ± 0.34, *p* = 0.027), as well as for the intial and middle 20 min of SS (0.22 ± 0.32 °C, *p* = 0.043, and 0.33 ± 0.33 °C, *p* = 0.012), while the difference for final part of the SS was not statistically significant (*p* = 0.266), see Figure 2. Mean core body temperature (with standard deviations) as determined with the MSR rectal sensor (red line and shaded area) and CORE (blue) sensor during Study I. (Figure 2b) Bland-Altman plot of the data in (Figure 2a). The dashed lines represent the limits of agreement (LoA) and bias for the entire period of exercise.

The corresponding Bland Altman plot is presented in Figure 2b. The limits of agreement for the whole workout were −0.38 °C and 0.72 °C, see Figure 2. Mean core body temperature (with standard deviations) as determined with the MSR rectal sensor (red line and shaded area) and CORE (blue) sensor during Study I. (Figure 2b) Bland-Altman plot of the data in (Figure 2a). The dashed lines represent the limits of agreement (LoA) and bias for the entire period of exercise.

Moreover, the analysis showed that the lowest range between LoA was for the warm-up period (−0.21 to 0.77 °C) and the largest for the last 20 min of SS (−0.64 to 0.56 °C).

Table 4 presents the temperatures at the end of each phase (warm-up; initial, middle, and final 20 min of SS), as well as the total change in temperature during the entire and each phase of the exercise measured with a CORE sensor and an MSR rectal sensor. The end temperature of the warm-up period and the initial 20 min of SS was significantly lower when measured with the CORE sensor as compared to the MSR rectal sensor (*p* = 0.016). The increase in temperature was significantly higher for the CORE sensor during the warm-up period (*p* = 0.012), while there was no statistically significant difference for the other phases of the exercise.

#### 3.2.2. Moderate-to-High Heat Load

Additional tests were performed to verify the validity of the CORE sensor at moderate-to-high heat load (Study II). Figure 3a presents the mean *T*_c_ of the participants, measured with an MSR rectal sensor and a CORE sensor. The difference in mean *T*_c_ measured with the MSR rectal sensor and the CORE sensor was statistically significant (−0.10 ± 0.38 °C, *p* < 0.001). The negative mean bias means that the CORE sensor underestimated the temperature from the rectal sensor. Moreover, data show that mean differences between devices were below a previously established threshold of 0.3 °C in 45% of all values for the entire exercise, see Table 5. 

The analysis was performed separately for each phase of the exercise: warm-up period, Ramp to RCP, every 20 min of SS, and cooling down. Wilcoxon signed-rank test showed that the difference in mean temperatures between the CORE and the MSR rectal sensor was statistically significant for all phases of the exercise, see Table 5.

The Bland Altman plot is presented in Figure 3b. The limits of agreement for the whole exercise were −0.62 °C and 0.59 °C, see Table 5. Moreover, the analysis showed that the lowest range between LoA was for the cooling down period (−0.68 to 0.33 °C) and the largest for the warm-up period (−0.75 to 0.42 °C).

Table 6 presents the temperatures at the end of each phase (warm-up; Ramp; initial, middle, and final 20 min of SS; cooling down), as well as the total temperature change during exercise as measured with the CORE sensor and the MSR rectal sensor. The end temperature of the warm-up period was significantly higher when measured with the CORE sensor in comparison with the MSR rectal sensor (*p* = 0.020). The increase in temperature was significantly higher for the CORE sensor during Ramp (*p* < 0.001), and cooling down period (*p* = 0.005), while for the other parts of the workout there was no statistically significant difference.

## 4. Discussion

The main purpose of the current investigation was to evaluate the reliability and validity of a novel device (CORE) that is claimed to estimate *T*_c_ accurately during indoor cycling under conditions of low-to-moderate heat load, as well as the validity of this same sensor at moderate-to-high heat load. The main findings were that the reliability of the CORE sensor was acceptable, with a non-significant mean bias between Trials 1a and 1b in Study I of only 0.02 °C. However, in comparison to the “gold standard” MSR rectal sensor, the *T*_rec_ indicated by the CORE sensor demonstrated poor agreement during cycling under conditions of both low-to-moderate and moderate-to-high heat load, with differences between the devices that were greater than the predefined acceptable level of ≤ 0.3 °C being associated with 45% and 51% of all values measured, respectively. These findings do not support the claim that the CORE sensor provides a valid measure of core body temperature.

### 4.1. Reliability 

In Study I, exercise-induced changes in *T*_c_ were similar between the repeated exercise trials (i.e., Trial 1a and Trial 1b). We observed a systematic bias of 0.02 ± 0.23 °C and LoA of −0.30 to +0.42 °C. Gant et al. [16], and Ruddock et al. [2] have published studies dealing with the reliability of ingestible pills. The means bias and LoA assessed in our study was lower compared to the mean bias of −0.07 ± 0.31 °C and LoA of ± 0.61 °C reported by Ruddock et al. [2]. Exercise intensity duration was similar as in our study, while the ambient temperature in the laboratory was higher (35 ± 0.2 °C vs 19.1 ± 0.6 °C). The mean bias reported by Gant et al. [16] was similar to our mean bias (0.01, and 0.02 ± 0.23 °C, respectively), while the LoA was lower compared to ours (± 0.23 °C, and −0.30 to + 0.42 °C). They assessed the reliability during intermittent running in a cool environment.

More detailed analysis showed that the mean bias between Trial 1a and Trial 1b was statistically significant during the warm-up period (0.25 ± 0.34 °C, *p* = 0.027), while during SS there was no statistically significant difference. According to the manufacturer’s instructions, the CORE sensor should be connected to a heart rate monitor during exercise and disconnected during rest to obtain the most accurate readings. However, the state of the heart rate connection cannot be changed during the measurement and therefore, the temperature values obtained during rest (at the beginning of the exercise) may not be accurate. These potentially inaccurate CORE temperature values at the beginning of the exercise can explain the statistically significant difference in mean bias between Trial 1a and Trial 1b during the warm-up period. In addition, this can explain the statistically different increase in temperature during the entire exercise for Trial 1a and Trial 1b. However, it has to be acknowledged that from a sports science perspective the initial 5 min of exercise (warm-up), when the core body temperatures are still below 38 °C, are less important in terms of training/performance.

To allow evaluation of the reliability of the CORE sensor, the participants had to exercise under the same conditions (i.e., environmental conditions and exercise intensity). During Study II, the laboratory temperature was high, raising the possibility that the body core could become too hot (above 39.5 °C), which would require a reduction in exercise intensity. Moreover, exercise under such hot conditions could result in inducing a certain degree of heat acclimation and thereby influence the participants’ subsequent responses. Therefore, sensor reliability could not be assessed in connection with Study II.

### 4.2. Validity

#### 4.2.1. Study I

The results of Study I show that a systematic bias between the temperature values obtained from two different sensors was evident throughout the protocol (0.23 ± 0.35 °C, *p* < 0.001), with the temperatures of the CORE sensor being systematically higher than those from the MSR rectal sensor, see Figure 2a. The range of differences in temperatures between devices was within the sum (±0.46 °C) of the measurement error provided by the manufacturers of each device (±0.2 °C for rectal sensor, and ±0.26 °C for CORE sensor) in 66% of all measured data points. Moreover, the mean difference between devices was below the criterion threshold of 0.3 °C in 51% of all measured data points, which is much lower compared to the percentage reported by Gosselin et al. (91%) [15]. Gosselin et al. tested the validity of the ingestible sensor during treadmill running in a hot environment (ambient temperature 38 °C).

A more detailed analysis showed that the mean bias in temperature between both devices was statistically significant and varied from around 0.22 ± 0.33 to 0.33 ± 0.33 °C across all phases, except for the last 20 min of SS. The observed systematic bias was higher than reported by Gosselin et al. [15] and Gant et al. [16] that compared the ingestible temperature sensor (pill) with the rectal sensor. They reported a mean bias ranging from 0.1 to 0.2 °C. Gant et al. assessed the validity of an ingestible temperature sensor during intermittent running in a cool environment. Interestingly, in this study, the temperature measured by the ingestible temperature sensor was systematically higher compared to the temperature from the rectal sensor, while in the study by Gosselin et al. the ingestible temperature sensor underestimated the temperature measured with the rectal sensor.

Despite the systematic difference in the temperature was observed between the CORE sensor and the MSR rectal sensor, the total temperature increase was, however, shown not to significantly differ between devices for the entire exercise, as well as each phase of exercise except the warm-up period (Table 4). Statistically significant different increases of temperature between both sensors during the warm-up period can be explained similarly as in Section 4.1. 

#### 4.2.2. Study II 

The results of Study II show that a systematic bias between the temperature values obtained from two different sensors was evident throughout the protocol (−0.10 ± 0.38 °C, *p* < 0.001). In contrast to Study I, the mean temperature obtained with the CORE sensor was lower compared to the values obtained from the MSR rectal sensor during the entire exercise.

The range of differences in temperatures between devices was within the sum (±0.46 °C) of the measurement error provided by the manufacturers of each device in 73% of all measured data points, which was a slightly higher percentage compared to Study I. Moreover, the mean difference between devices was below the criterion threshold of 0.3 °C in 45% of all measured data points, which is much lower compared to the percentage reported by Gosselin et al. (91%) [15].

A more detailed analysis showed that the mean bias was not constant for all phases of the exercise. At the beginning and the end of the exercise bout, the CORE sensors underestimated the temperature obtained with MSR rectal sensor, while in the middle (SS from 15 to 35 min) the CORE sensor overestimated the temperature obtained with MSR rectal sensor.

Although the *T*_rec_ is the preferred and recommended method of one of the governing bodies—National Athletic Trainers’ Association for assessing core body temperature [24], athletes and coaches use a variety of devices to measure temperature which is less invasive compared to the rectal sensor. Compared to the data published by Ganio et al. [17], the CORE sensor has proven to be more accurate than other non-invasive devices (i.e., devices to assess forehead, oral, temporal, aural, and axillary) used in sports. Nevertheless, the studies showed [15,16,17] that the ingestible temperature sensors are still more valid compared to the CORE sensor, but they are not entirely non-invasive and associated with high costs.

The results of the present study must be interpreted with the following limitations in mind. We only tested continuous exercise, steady-state cycling. The main reason is that, as stated by Taylor et al. [8], the rectal temperature is perfectly acceptable during steady states while inadequate in certain dynamic phases. Therefore, the sensor response during intermittent exercise, for example, remains unknown. Moreover, the exercise was not performed in either very cold or very hot (above 30 °C) environmental conditions. In addition, only males were included here, primarily because the temperature changes associated with the menstrual cycle [25] could have influenced our evaluation of reliability. Clearly, this limitation should be kept in mind when interpreting data on women obtained with the CORE sensor. Accordingly, we utilized the *T*_rec_ as the *T*_c_ reference value. As reported previously, *T*_rec_, gastrointestinal and esophagus temperatures are comparable when changes in the core temperature are small and/or gradual [26], whereas during the rapid changes only *T*_rec_ and gastrointestinal temperature correlate well [27]. Therefore, although *T*_rec_ does, in fact, reflect the actual *T*_c_ in most situations, in some cases, this value may be an under or -overestimation [8]_._ This potential limitation should be taken into consideration when interpreting our present findings and in future studies measurement of *T*_c_ at multiple sites could provide an even better reference value.

## 5. Conclusions

Our findings indicate clearly that measurements of the core body provided by the CORE sensor are acceptably reliable, since the mean bias between repeated trials did not differ significantly. However, mean differences between these measurements and those provided by the MSR rectal sensor were greater than the predefined acceptable threshold of <0.3 °C in connection with approximately 50% of all the measurements we performed. Accordingly, our present findings do not support the claim that the CORE sensor provides valid measurements of core body temperature in male cyclists, therefore athletes and coaches should interpret such measurements with caution. In particular, care should be taken when assessing/monitoring higher *T*_c_ (above 39.5 °C) associated with heat-related medical problems, since the CORE sensor underestimates such elevated core body temperatures.

## Figures and Tables

**Figure 1 sensors-21-05932-f001:**
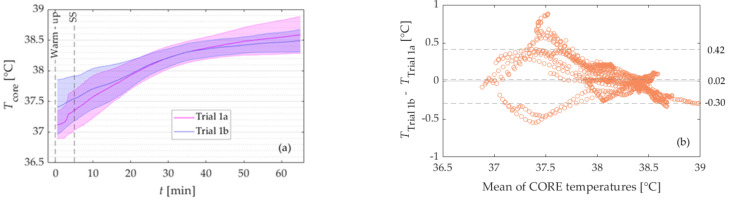
(**a**) Mean core body temperature as determined with the CORE sensor during Trial 1a (pink) and Trial 1b (violet) of Study I. The shaded areas represent the corresponding standard deviations. (**b**) Bland-Altman plot of the data in a). The dashed lines represent limits of agreement (LoA) and bias for the entire trials.

**Figure 2 sensors-21-05932-f002:**
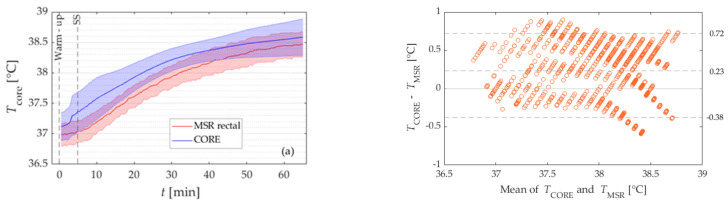
Mean core body temperature (with standard deviations) as determined with the MSR rectal sensor (red line and shaded area) and CORE (blue) sensor during Study I. (**b**) Bland-Altman plot of the data in (**a**). The dashed lines represent the limits of agreement (LoA) and bias for the entire period of exercise.

**Figure 3 sensors-21-05932-f003:**
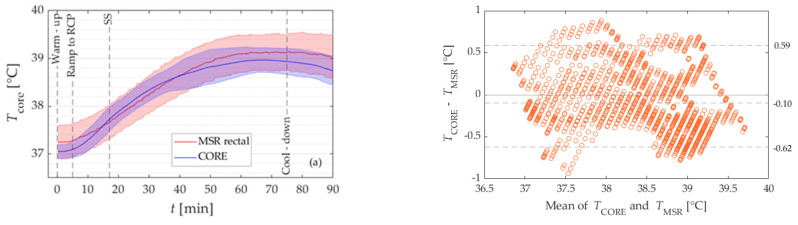
The mean temperature (and standard deviation, the shaded areas) during the entire exercise protocol execured in connection with Study II as determined with the MSR rectal sensor (red) and CORE sensor (blue) sensor. (**b**) Bland-Altman plot of the data in (**a**). The dashed lines represent the limits of agreement (LoA) and bias for the entire exercise.

**Table 1 sensors-21-05932-t001:** Mean core body temperatures as measured with the CORE sensor during Trials 1a (first column) and 1b (second column) for the entire period of exercise, the 5-min warm-up, and 20-min intervals of the steady-state (SS) exercise. In addition, the third and fifth columns document the mean bias and limits of agreement (LoA), respectively.

Periods of Exercise	CORE Trial 1a [°C]	CORE Trial 1b [°C]	Bias [°C]	*p*-Value	LoA [°C]
Entire	38.11 ± 0.48	38.13 ± 0.39	0.02 ± 0.23	0.622	−0.30 to + 0.42
Warm-up [0–5 min]	37.23 ± 0.25	37.48 ± 0.40	0.25 ± 0.34 *	0.027	−0.34 to + 0.86
**SS cycling**					
Initial 20 min	37.74 ± 0.34	37.82 ± 0.31	0.08 ± 0.28	0.266	−0.45 to + 0.52
20–40 min	38.29 ± 0.20	38.28 ± 0.16	−0.01 ± 0.10	0.967	−0.16 to + 0.21
Final 20 min	38.51 ± 0.25	38.44 ± 0.15	−0.07 ± 0.14	0.233	−0.28 to + 0.18

* significantly different mean temperature during Trials 1a and 1b 2 (*p* < 0.05).

**Table 2 sensors-21-05932-t002:** Mean core body temperatures indicated by the CORE sensor during the final 30 sec of Trial 1a (*T*_end1a_) and Trial 1b (*T*_end1b_), as well as of each individual phase. In addition, the fourth and fifth columns document the corresponding total increase in temperature.

Periods of Exercise	*T*_end1a_ [°C]	*T*_end1b_ [°C]	*p* Value	Δ*T*_1a_ [°C]	Δ*T*_1b_ [°C]	*p* Value
Entire	38.58 ± 0.30	38.49 ± 0.18	0.204	1.46 ± 0.28	1.09 ± 0.44 *	0.021
Warm-up [0–5 min]	37.36 ± 0.33	37.55 ± 0.37	0.064	0.24 ± 0.25	0.15 ± 0.13	0.339
**SS cycling**						
Initial 20 min	38.09 ± 0.20	38.10 ± 0.17	0.622	0.72 ± 0.19	0.55 ± 0.21 *	0.043
20–40 min	38.43 ± 0.20	38.39 ± 0.13	0.349	0.34 ± 0.12	0.29 ± 0.13	0.064
Final 20 min	38.58 ± 0.30	38.49 ± 0.18	0.204	0.15	0.10 ± 0.09	0.204

* significantly different from Trial 1a (*p* < 0.05).

**Table 3 sensors-21-05932-t003:** Mean core body temperatures as determined with the CORE sensor (first column) and MSR rectal sensor (second column) during the entire Study I, as well as during the corresponding 5-min warm-up and 20-min intervals of steady-state (SS) exercise. In addition, the third column presents the mean bias, the fifth the limits of agreement (LoA), and the sixth the proportion of measurements that were below the pre-defined threshold of 0.3 °C.

Periods of Exercise	CORE [°C]	MSR [°C]	Bias [°C]	*p*	LoA	≤0.3 °C [%]
entire	38.11 ± 0.48	37.88 ± 0.52	0.23 ± 0.35 *	<0.001	−0.38 to + 0.72	51
Warm-up [0–5 min]	37.23 ± 0.25	37.01 ± 0.18	0.22 ± 0.32 *	0.043	−0.21 to + 0.77	64
**SS cycling**						
Initial 20 min	37.74 ± 0.34	37.40 ± 0.29	0.33 ± 0.33 *	0.012	−0.19 to + 0.80	45
20–40 min	38.29 ± 0.20	38.06 ± 0.23	0.24 ± 0.31 *	0.027	−0.38 to + 0.64	37
Final 20 min	38.51 ± 0.25	38.39 ± 0.21	0.13 ± 0.37	0.266	−0.64 to + 0.56	43

* significantly different mean temperature for CORE and MSR rectal sensor (*p* < 0.05).

**Table 4 sensors-21-05932-t004:** Mean temperatures during the last 30 sec of Study I, as well as of the corresponding phases of exercise, as determined with the CORE sensor (*T*_endCORE_) and MSR rectal sensor (*T*_endMSR_). The fourth and fifth columns report the increase in total temperature increase during each period.

Periods of Exercise	*T*_endCORE_ [°C]	*T*_endMSR_ [°C]	*p* Value	Δ*T*_CORE_ [°C]	Δ*T*_MSR_ [°C]	*p* Value
entire	38.58 ± 0.30	38.47 ± 0.21	0.380	1.46 ± 0.28	1.48 ± 0.29	0.677
Warm-up [0–5 min]	37.36 ± 0.33	37.04 ± 0.18 *	0.016	0.24 ± 0.25	0.06 ± 0.05 *	0.012
**SS cycling**						
Initial 20 min	38.09 ± 0.20	37.79 ± 0.19 *	0.016	0.72 ± 0.19	0.75 ± 0.20	0.910
20–40 min	38.43 ± 0.20	38.27 ± 0.21	0.151	0.34 ± 0.12	0.47 ± 0.14	0.064
Final 20 min	38.58 ± 0.30	38.47 ± 0.21	0.380	0.16 ± 0.15	0.20 ± 0.10	0.204

* significantly different for CORE and MSR rectal sensor (*p* < 0.05).

**Table 5 sensors-21-05932-t005:** Mean core body temperatures indicated by the CORE (first column) and MSR rectal (second column) sensors for the protocol during Study II, as well as the individual phases of this exercise. In addition, the third column represents the mean bias, the fifth limits of agreement (LoA), and the sixth the proportion of all individual measurements ofr which the LoA was less than the pre-defined threshold of 0.3°C.

Periods of Exercise	CORE [°C]	MSR [°C]	Bias [°C]	*p* Value	LoA	≤0.3 °C [%]
Entire	38.41 ± 0.68	38.51 ± 0.77	−0.10 ± 0.38 *	<0.001	−0.62 to +0.59	45
Warm-up [0–5 min]	37.06 ± 0.14	37.29 ± 0.34	−0.22 ± 0.30 *	<0.001	−0.75 to +0.42	62
Ramp [5–15 min] SS [15–35 min]	37.31 ± 0.26	37.44 ± 0.36	−0.13 ± 0.30 *	<0.001	−0.63 to +0.42	60
	38.11 ± 0.34	38.02 ± 0.44	0.09 ± 0.35 *	0.031	−0.35 to +0.72	62
SS [35–55 min]	38.72 ± 0.33	38.79 ± 0.43	−0.06 ± 0.45 *	<0.001	−0.63 to +0.67	33
SS [55–75 min]	38.94 ± 0.27	39.12 ± 0.36	−0.18 ± 0.34 *	<0.001	−0.61 to +0.45	35
Cool-down [75–90 min]	38.86 ± 0.28	39.12 ± 0.40	−0.22 ± 0.30 *	<0.001	−0.68 to +0.33	37

* significantly different mean temperature for CORE and MSR rectal sensor (*p* < 0.05).

**Table 6 sensors-21-05932-t006:** Mean temperatures during the final 30 sec of the entire exercise protocol in Study II, as well as of each individual phase of this exercise, as measured with the CORE (*T*_endCORE_) and MSR rectal (*T*_endMSR_) sensors. In addition, the fourth and fifth columns indicate the total increase in temperature during each period.

Periods of Exercise	*T*_endCORE_ [°C]	*T*_endMSR_ [°C]	*p*	Δ*T*_CORE_ [°C]	Δ*T*_MSR_ [°C]	*p*
Entire	38.74 ± 0.30	39.06 ± 0.46	0.003 *	2.04 ± 0.27	2.05 ± 0.34	0.906
Warm-up [0–5 min]	37.31 ± 0.34	37.10 ± 0.16	0.020 *	0.07 ± 0.07	0.05 ± 0.05	0.366
Ramp [5–15 min]	37.60 ± 0.26	37.59 ± 0.35	1	0.50 ± 0.15	0.28 ± 0.08	<0.001 *
SS [15–35 min]	38.51 ± 0.24	38.46 ± 0.41	0.839	0.90 ± 0.18	0.87 ± 0.21	0.825
SS [35–55 min]	38.88 ± 0.33	39.03 ± 0.40	0.244	0.37 ± 0.20	0.57 ± 0.14	0.005 *
SS [55–75 min]	38.94 ± 0.27	39.15 ± 0.37	0.040 *	0.06 ± 0.25	0.12 ± 0.28	0.273
Cool-down [75–90 min]	38.74 ± 0.30	39.06 ± 0.46	0.003 *	−0.19 ± 0.21	−0.09 ± 0.26	0.017 *

* significantly different for CORE and MSR rectal sensor (*p* < 0.05).

## Data Availability

The data presented in this study are available on request from the corresponding author providing the access does not interfere with the conditions provided by the ethics committee.

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
