# Peer review of "Reliability and Validity of the CORE Sensor to Assess Core Body Temperature during Cycling Exercise"

_sensors, 2021, doi:10.3390/s21175932_

Round 1

Reviewer 1 Report

The reviewer applauds the researchers for this timely study relating to assessing the efficacy of the Core sensor.

The reviewer has several questions:

1) "The accuracy of 173 the CORE device is described by the manufacturer to be ± 0.26°C." Was the criterion for an accuracy of 0.3 C (mentioned in the abstract) taken from the 0.26 value?

2) What were the inclusion and exclusion criteria for this study? Why were  24 male athletes selected? Given the differences in physiology between men and women, the reviewer strongly suggests the authors perform the similar study on female cohorts to atleast assess the efficacy of the sensor in this population. If female athletes could not be consented, please mention. Else, to solely conclude that the sensor is not accurate using 24 healthy male individuals is not sufficient.

Reviewer 2 Report

The present manuscript focused on the reliability and validity of the wearable sensor named “CORE Sensor” by two different study projects conducted in the experimental chamber. The information is very important; however, I have several concerns.

Introduction

Page 2, line 56. Although rectal temperature is a reliable indicator to assess core body temperature, the value could not be gold standard for core body temperature. As the review by Dr Nigel Taylor, the value shows sometimes higher than the other values repoted to represent core body temperature. When subjects conducted exercise using legs such as running and cycling, rectal temperature tends to show greater value than the other core parts. Even in this condition, rectal temperature could be used as core temperature. But, when we develop a tool aiming to measure core body temperature, I think that we could not evaluate the accuracy or reliability only by the identity of the value with rectal temperature. This is because core temperature is different among the parts of the core body. I recommend to check the linearity between changes of rectal temperature and the value of CORE sensor during exercise.

Page2, line 80-89. The authors conducted similar analyses which were conducted to assess the reliability of the pill-type sensor. However, the authors did not show any information about the principle of CORE sensor, which is confusing. The manufacturer may not open the information; however, even in the situation, I think that the authors needed to consider the disturbing factors such as skin temperature or sweating.

Methods

I could not understand why the reliability could be verified by the data obtained in Study I. The authors consider that rectal temperature could be used as a reference of core body temperature. I may misunderstand; however, it is easiest way to evaluate if CORE sensor shows the similar value at the given rectal temperature in the same subject (intra-individual difference) and among subjects (inter-individual difference).

Study II was conducted in a mid-hot environment. On the contrary, Study I was conducted in a cool and dry condition. The sensor may be influenced by skin temperature and sweating. In addition, the influence decreased the repeatability even during one exercise experiment. Clearer rationale would be needed why the authors conducted the two different study projects.

Reviewer 3 Report

This study was conducted to compare the validity and reliability of the CORE sensor to that of a rectal sensor under various laboratory conditions and to examine the reproducibility of values obtained with the CORE sensor during exercise under the same conditions on two separate occasions. The results indicated that measurements of the core body provided by the CORE sensor are acceptably reliable. However, the results do not support the manufacturer´s claim that the CORE sensor provides a valid measure of core body temperature. Overall, this research topic is interesting, and the study methods are appropriate. There are some issues to consider for the authors.

  1. Please show the information of the CORE sensor clearly, such as a thermistor or infrared sensor. In addition, according to the manufacturer’s instructions, the CORE sensor requires a thermal equilibration time to avoid unstable body temperature readings, which may be one of the factors affecting reliability.

  1. The inclusion and exclusion criteria of study participants are unclear to assess the generalizability of the study results.

  1. Because the CORE sensor was positioned on the torso/chest, the accuracy of temperature might be affected by sweating and hairy skin at the measurement site. It would be better to describe more about the potential influencing factors of body temperatures in this study.

Round 2

Reviewer 1 Report

I thank the authors for making the requested changes and look forward to this work being published.

Reviewer 3 Report

The authors have improved the manuscript following the revision.